# Does Seasonal Influenza Related Hospital Occupancy Surge Impact Hospital Staff Sickness Presenteeism and Productivity Costs?

**DOI:** 10.3390/ijerph19020769

**Published:** 2022-01-11

**Authors:** Juliana Nga Man Lui, Ellie Bostwick Andres, Janice Mary Johnston

**Affiliations:** School of Public Health, Li Ka Shing Faculty of Medicine, The University of Hong Kong, G/F, Patrick Manson Building (North Wing), 7 Sassoon Road, Pokfulam, Hong Kong SAR, China; julianal@connect.hku.hk (J.N.M.L.); eandres@hku.hk (E.B.A.)

**Keywords:** presenteeism, productivity, nurses, influenza, hospital occupancy

## Abstract

Background—The workload of public hospital staff is heightened during seasonal influenza surges in hospitals serving densely populated cities. Such work environments may subject staff to increased risk of sickness presenteeism. Presenteeism is detrimental to nurses’ health and may lead to downstream productivity loss, resulting in financial costs for hospital organizations. Aims—This study aims to quantify how seasonal influenza hospital occupancy surge impacts nurses’ sickness presenteeism and related productivity costs in high-intensity inpatient metropolitan hospitals. Methods—Full-time nurses in three Hong Kong acute-care hospitals were surveyed. Generalized estimating equations (GEE) was applied to account for clustering in small number of hospitals. Results—A total of 71.3% of nurses reported two or more presenteeism events last year. A 6.8% increase in hospital inpatient occupancy rate was associated with an increase of 19% (1.19, 95% CI: 1.06–1.34) in nurse presenteeism. Presenteeism productivity loss costs between nurses working healthy (USD1983) and worked sick (USD 2008) were not significantly different, while sick leave costs were highest (USD 2703). Conclusion—Presenteeism prevalence is high amongst acute-care hospital nurses and workload increase during influenza flu surge significantly heightened nurse sickness presenteeism. Annual presenteeism productivity loss costs in this study of USD 24,096 were one of the highest reported worldwide. Productivity loss was also considerably high regardless of nurses’ health states, pointing towards other potential risk factors at play. When scheduling nurses to tackle flu surge, managers may want to consider impaired productivity due to staff presenteeism. Further longitudinal research is essential in identifying management modifiable risk factors that impact nurse presenteeism and impairing downstream productivity loss.

## 1. Introduction

Presenteeism is a behaviour in which an employee is physically present at work with reduced performance due to illness or other reasons [1]. This type of behaviour may bring about substantial economic losses, detrimental consequences to employee health and well-being, and impede upon healthcare workforce sustainability [2,3,4]. A study in Sweden reported that nurses were 2.7 to 4.3 times more likely to experience presenteeism than office managers [3]. Presenteeism amongst hospital staff, nurses in particular, is also associated with adverse patient outcomes such as increased risk of nurse-to-patient disease transmission and reduced quality of care [5,6].

### 1.1. Seasonal Influenza Flu Surge and Staff Presenteeism in Metropolitan Hospitals

Public hospital staff work in high-intensity inpatient settings, particularly in hospitals located in a densely populated metropolitan city as Hong Kong, are characterized by over-occupancy and manpower shortages. Hong Kong’s public hospitals provide 90 percent of all inpatient bed-days. Significant increases in bed occupancy rates to over 100% were observed in Hong Kong public hospitals during the bimodal summer and winter peak periods of seasonal influenza activity [7]. Such work environments may subject hospital staff to increased risk of sickness presenteeism [8].

### 1.2. Hong Kong Hospital Cluster Division and Nurse Workforce

To facilitate healthcare resources allocation, the public hospital service management is divided into 7 geographical clusters, known as Hong Kong East (HKE), Hong Kong West (HKW), Kowloon East (KEC), Kowloon West (KWC), Kowloon Central (KCC), New Territories East (NTE) and New Territories West (NTW). Currently, each cluster is responsible for annual budget plans to facilitate resource allocation based on actual service utilization by patients in their designated catchment service area [9]. Resources are allocated to respective clusters based on historical workload. Nurses in the Kowloon clusters (KEC, KWC and KCC) face the highest service demands (total number of inpatient days) with lowest manpower (nurse ratio to inpatient days) as compared to other clusters [10,11] (Figure 1).

Nurses make up 57% of Hong Kong healthcare manpower and the number of registered and enrolled nurses (RNs and ENs respectively) increased by 45% between 2008 to 2017 [12], with 40,505 RNs and 13,726 ENs in Hong Kong in 2017 [13]. Despite the increase in nurse manpower, the Government estimated in 2017–2018 that there is a shortage of about 400 nurses in the public hospital sector [14].

### 1.3. Productivity Loss Costs, Influenza and Staff Presenteeism

Productivity loss costs are indirect costs that arise from two types of employee work attendance behaviour, absenteeism and presenteeism [15]. Productivity loss costs related to presenteeism were found to outweigh absenteeism costs [2]. A study in United States reported that migraine-related presenteeism costs exceeded absenteeism costs by USD 3 million [16]. Presenteeism costs that were related 10 common health conditions accounted for 61% of total average expenditures amongst US employers [17]. Employee chronic health condition costs in Dow Chemical Company related to presenteeism were highest (USD 6721) as compared to absenteeism (USD 661) and direct medical costs (USD 2278) [18]. In high workload settings, especially service organizations such as hospitals, staff are more prone to exhibit sickness presenteeism behavior and incur related productivity costs [3].

Employees with influenza-related acute respiratory infections are significantly associated with workplace productivity loss [19]. In Wormer’s study, regardless of vaccination status and virus strain subtypes, those exhibiting presenteeism with an influenza infection had a higher mean productivity loss of 67–74%, as compared to 58–59% of productivity loss amongst employees with non-influenza related acute respiratory infections [19]. Another modelling study predicted that it was cost-effective to vaccinate healthcare staffs against influenza, with highest cost savings from avoided presenteeism productivity losses of USD 10,303 [20].

Our healthcare workforce is also prone to influenza-related sickness presenteeism while attending to the high work demands of flu surge presenteeism related productivity loss poses great challenge to hospital managers in staff scheduling of nurses to satisfy demands during high occupancy surge periods.

Most existing nurse presenteeism studies examine presenteeism prevalence and its related risk factors amongst nurses, while little research has been carried out to explore the productivity loss costs and impact of nurse sickness presenteeism [21]. There is reported association between presenteeism and productivity [22,23], and between influenza infection and productivity loss (absenteeism/presenteeism combined) [19,24], and flu-related productivity loss costs were reported amongst Turkish white-collar professionals [25]. However, the above-mentioned studies were not carried out in the healthcare sector. Moreover, the relationship between influenza flu surge work demands (proxied by bed occupancy rates), nurse presenteeism and financial costs of influenza related productivity loss has not been studied.

It was shown that efficient strategic healthcare human resources would help reduce presenteeism and promote a sustainable, healthy workforce for higher organization performance in the long term [26]. It is crucial for hospital managers to understand the relationship between increased occupancy rates during influenza season, flu-related presenteeism and its related productivity losses amongst nurses so that they can factor in the health and well-being of hospital staff when budgeting their actual productivity output during manpower allocation exercises. Evidence is also needed for hospital managers to evaluate the financial impact that flu-related presenteeism productivity loss brings about to the organization.

This study aims to quantify how increased workload during seasonal influenza, as proxied by hospital occupancy surge, impacts staff nurses’ sickness presenteeism and related productivity costs in a densely populated metropolitan city inpatient hospital setting.

## 2. Materials and Methods

### 2.1. Sample Frame

The study sampling frame consists of all full-time nurses employed in three acute care hospitals (H1, H2 and H3) serving the most densely populated Kowloon hospital clusters in Hong Kong with the lowest nurse ratio to inpatient days (Figure 1) (N_H1_ = 2145, N_H2_ = 1367 and N_H3_ = 1145). The sampled hospitals provide a full range of acute care in-patient and outpatient specialty and sub-specialty services. Hong Kong is highly geographically accessible due to its small land mass, thus cross-district hospital utilization is not uncommon. The selected hospitals have the highest inpatient occupancy and cross-district hospital utilization rates in Hong Kong. H1 and H2 are both major tertiary level hospitals serving their cluster district, where H2 is also a neurosurgical and antenatal diagnosis referral center and has high level of cross-cluster patients [27]. H3 is a quaternary level hospital and designated trauma center which provides high intensity level of patient care. The three hospitals are distinct in terms of size (number of beds: H1 = 1186, H2 = 1433 and H3 = 1932), level of care, organizational culture, religious affiliation and target population, thus providing a more generalizable sample that considers the variance in other immeasurable aspects of workload of distinct hospitals (intensity of care, organizational culture, administrative and management activity levels) [28,29].

### 2.2. Survey Administration and Collection Procedure

Survey packets containing: (1) a study information sheet, (2) a survey labelled with a unique identification number (UIN), (3) a self-sealed return envelope, and (4) a small incentive (valued less than 1 USD) were prepared and delivered to the nurses at their primary working location.

To ensure subject confidentiality, a unique identification number (UIN) was generated for each respondent based on nurse roster information prepared by the central nursing department at each hospital. The first 11 digits of the UIN were alpha numerically encoded for the respective hospital, department, ward and rank of nurse, while the last five digits comprised a random number computer generated for each individual. The file linking respondents’ identity and UIN was only available to the chief research assistant who generated the UIN labelled surveys and packed them in the addressed envelopes. All other investigators were blinded to individual level data.

Research staff distributed and subsequently collected the completed surveys (sealed in return envelopes) by ward in each hospital. To achieve the highest possible response rate, non-responder follow-up surveys were sent one month post initial contact. Completed surveys were collected three times during the whole cross-sectional surveyed period (Figure 2).

### 2.3. Measures

Occupancy rate is selected as a proxy for hospital workload in this study as it objectively reflects seasonal fluctuations in hospital level nurse workload. Hong Kong is geographically located in the subtropical climate region, where influenza exhibits a biannual seasonality during winter and summer months [30]. The cross-sectional survey period was designed to capture inpatient occupancy peak rates between late February and early May in 2017 to test the effects of different work demand stress levels on nurse presenteeism behavior. Inpatient bed occupancy rate is the ratio of the number of inpatients to the number of inpatient beds (including temporary beds). Daily inpatient occupancy rates varied on average by 28% in the three hospitals across the 2-month collection period (Figure 2). The ideal method of assigning daily occupancy rates to each survey according to which date the survey was completed is restrained by administrative difficulties. Therefore, an average of daily inpatient occupancy rates during the three collection periods was assigned to when the survey was collected.

The primary outcome, presenteeism frequency, was measured using a single numerical item “How many times during the last year have you gone to work when you should have been on sick leave due to your health condition?” Responses were rated on a four-point scale (1—never, 2—once, 3—two to five times, 4—more than five times). This measure was developed by Aronsson [3], with a reported test-retest reliability of 0.58 for a 12-month recall period [31], and commonly used in the field [32,33]. Longer recall periods (1 year) are more appropriate to measure sickness presenteeism, which is considered a type of long-term persistent work behavior [34]. Studies have reported that shorter recall periods (2 months) may underestimate the prevalence of presenteeism in sampled population [35].”

Self-reported productivity was measured using the item “how would you rate your overall job performance on the days you worked during the past 4 weeks (28 days)?” adopted from the World Health Organization’s Heath and Work Performance Questionnaire (HPQ) short form [36], rated on a 10-point visual analogue scale (0—worst performance, 10—best performance). In contrast to sickness presenteeism frequency, a shorter recall period (one to four weeks) is recommended for productivity loss estimates [37]. For ease of monthly productivity loss costing calculations, recall period of four weeks is selected in this study.

Sickness absenteeism was measured using two items “In the past 4 weeks (28 days), how many days did you miss an entire work day because of problems with your physical or mental health?” and “miss part of a work day because of problems with your physical or mental health?” from the HPQ measure.

General health was measured using a single item “compared with people of your age, do you consider that your health condition is”, rated on a 5-point Likert scale (1—very poor, 5—very good).

### 2.4. Data Analyses

#### 2.4.1. Presenteeism Prevalence

Presenteeism prevalence in nurses were presented in percentage-based tables in which comparisons were made by their belonging hospital.

#### 2.4.2. Association between Presenteeism and Workload

Generalized estimating equations (GEE) model, which accounts for clustering nature of our data, was used to evaluate the association between nurse presenteeism and workload. Since inflated Type I error rates may occur with a small number of clusters, GEE models are preferred over multilevel hierarchical logistic regression (HLM) models, as they allow for small-sample bias corrections to address this problem [38,39]. The primary outcome—presenteeism frequency was dichotomized (0—≤ 1 presenteeism event; 1—≥ 2 presenteeism events) for the purpose of logistic regression [40].

To account for intra-hospital inpatient occupancy fluctuations within the two-month collection period [41], average periodic inpatient occupancy rates were calculated and converted to z-scores. Average periodic occupancy rates (one of three collection timepoints (Figure 2)) were assigned to each completed survey based on collected time. Hospital occupancy rates for H3 for the third collection final week (5/5–5/13) were not available, as hospital occupancy rates are only publicly available for dates 20 December 2016–5 May 2017 [41]. The effect for this study is thought to be minimal since the week consisted of less than 30 returned surveys.

Covariates considered in the regression model included age, rank, general health, absenteeism, primary working location and working schedule (shift schedule or 9 am–5 pm) were selected a priori based on existing literature [34].

In alignment with the aim of this study, only inpatient ward nurses were included in the analysis, nurses who are in senior management or mainly work at outpatient clinics were excluded in the following analysis so to measure the effect of workload on presenteeism in inpatient ward nurses only.

Missing data were handled using multiple imputation with chained equations (MICE) [42], incomplete data sets were processed with a maximum of five iterations for ten imputations. Estimates and standard errors of complete-data GEE analyses performed on the imputed datasets were pooled together according to Rubin’s rule [43].

R statistical software version 3.4.2 was used for the statistical analyses. “Gee” package was used for the GEE regression modelling, “BCgee” package was used for small-sample bias correction, “mice” package was used for multiple imputation analysis on missing data, “norm” package was used to pool imputed datasets according to Rubin’s rule.

#### 2.4.3. Productivity Costing

Monthly estimates on productivity loss of nurses in different health states were calculated using the human capital method (HCM) [44], which discounts the employee’s salary by self-reported presenteeism productivity loss and hours. The self-rated productivity item with shorter recall period (within past four weeks) was selected for optimal presenteeism work impact estimate analysis [37].

Nurses were first categorized into three different health states (healthy, sickness presenteeism, took sick leave). Those categorized as “healthy” did not exhibit sickness presenteeism in the past year or took any sick leave in in the past month. Those categorized as “sickness presenteeism” worked sick in the past year but did not take sick leave in the past month. Those categorized as “took sick leave” took at least one day of sick leave regardless of working sick or not in past year.

Monthly productivity loss costs for nurses whose health states were categorized as healthy and sickness presenteeism were calculated by discounting their monthly median salary (USD 6083) by self-rated job performance loss percentage in last month. Monthly productivity loss costs for nurses who took sick leave were calculated by discounting their salary on their remaining days of work by self-rated job performance loss percentage in last month. Detailed methods health state categorization and productivity loss costs calculation are listed in Table 1.

Most existing studies compare employee presenteeism on-the job productivity loss costs directly to absenteeism sick leave costs [45,46]. This study takes a more comprehensive approach by considering both sick leave costs and on-the-job productivity loss costs (for remaining work days of the month) for nurses’ absenteeism in calculating monthly productivity loss costs.

Monthly productivity loss was presented in terms of percentage scores and converted to monthly productivity loss costs in terms of US dollars (1 USD = 7.8 HKD). A Chi-squared test was used to test if there is significant difference between the monthly productivity costs across different health states.

## 3. Results

### 3.1. Nurse Presenteeism Prevalence

Characteristics of nurse participants by hospital are presented in Table 2. Nurse characteristics differed significantly between hospitals except for participation by rank. Presenteeism prevalence was 71.3 percent in the sampled nurses, where respondents rated two or more presenteeism events in the past year (Table 3). Average self-rated productivity score of nurses was 6.67 out of 10.

### 3.2. Association between Workload and Nurse Presenteeism

All hospitals reported high occupancy rates during the survey period, with two above 100 percent occupancy for the collection period. Inter-hospital periodic occupancy rates ranged from 93.5 percent for H3 to 118.3 percent for H2 with a difference of 24.8 percent, while intra-hospital periodic occupancy rates ranged between 3.5–14.9 percent between the initial and non-responder follow-up surveys.

GEE regression results showed a 6.8 percent increase (1 standard deviation) in inpatient occupancy rate was associated with an increase of 19 percent (OR occupancy rates = 1.19; 95% CI: 1.05–1.35) in presenteeism (≥2 events in past year) (Table 4). Conventional logistic regression results were presented alongside with GEE results for reader’s reference. Full parameter estimates of both models are also available in Appendix A.

### 3.3. Productivity Lost Costs of Nurses in Different Health States

Self-rated productivity among nurses who were healthy, exhibited sickness presenteeism and those who took sick leave were not significantly different across hospitals (*p* = 0.85) (Table 5). Nurses who took sick leave reported a productivity loss costs of USD 2081 per nurse (including both on-the-job productivity loss costs and sick leave costs). On-the-job productivity loss costs were not significantly different between those who worked healthy (USD 1983) and exhibited presenteeism (≥2 presenteeism events in past year) (USD 2008) (Table 5). When factoring in sick leave costs, on-the-job productivity loss costs of those who worked healthy (USD 1983) and worked while sick (USD 2008) were significantly different from those who took sick leave (USD 2081) (*p* < 0.001). Sick leave costs comprise 23 percent (overall sick leave costs/ (sick leave + productivity loss costs): USD 622/USD 2703) of total labor costs.

## 4. Discussion

### 4.1. Workload and Presenteeism

Nurse presenteeism prevalence reported in our study of 71.3% is comparatively higher than that reported in studies performed in Western countries (e.g., US, Italy, Netherlands) which ranged between 50–62% [31,47,48,49]. This may be due to the selected sampling time period and setting, covering the highest workload periods of the year during seasonal influenza flu surge in acute care hospitals serving the most densely populated areas in a metropolitan city.

Objective workload measure (inpatient occupancy rate) has been commonly used in patient outcome studies [50,51]. Few studies have considered the association between hospital workload with nurse outcomes, such as burnout, job satisfaction and mental disorders but not with nurse work attendance behavior [52,53]. This study quantified workload using an objective measure (inpatient occupancy rates) and showed an increase in workload associates with higher risk of employees exhibiting presenteeism. A Nordic study suggested that workload and low staffing may induce high stress levels and associates with frequent sickness presenteeism occurrences in elderly care assistant healthcare workers [54]. Demerouti further confirmed the causal relationship between workload, burnout and presenteeism in a sample of Netherland nurses [31].

### 4.2. Hong Kong Nurse Presenteeism Productivity Costs Highest Worldwide

On-the-job productivity loss costs were slightly higher amongst nurses who experienced presenteeism than those who worked healthy, but lower than those who took sick leave. The annual on-the-job productivity costs for nurses exhibiting presenteeism in our sample were USD 24,096 (USD 2008 × 12 months), 55 percent higher than the highest reported cost (range: USD 2000–USD 15,541) [47,48,55] amongst healthcare employees in other studies [34].

There is little evidence in existing literature on flu-surge related presenteeism and absenteeism on-the-job productivity loss costs amongst healthcare professionals. In China, a nation-wide survey study reported that the nurse presenteeism related productivity loss (non-specific to flu) annually costs up to 0.7 billion USD (4.38 Yuan) [56]. On the other hand, studies on non-healthcare professionals indicate that flu-related productivity loss costs (both presenteeism and absenteeism) are significant. Tomonaga reported absenteeism related annual sick leave costs amongst Swiss nationals to be 112 million USD (103 million Francs) in 2017 but on-the-job productivity loss costs were not considered [24]. Whereas productivity loss (absenteeism and presenteeism combined) of Turkish white-collar workers who experienced common cold and flu cost 25,352 USD (455,641 Liras) per employee each season [25].

Our results concur with Rantanen’s study that found on-the-job productivity loss costs associated with presenteeism are lower than those of absenteeism for healthcare workers [55]. However, nationwide costing studies have reported that long term employee medical costs associated with sickness presenteeism may be higher than short term direct medical costs paid by employers for sick leave [15,17]. Demerouti also reported non-monetary negative impacts of presenteeism on organizations, such as long-term downstream health issues and increased stress and burnout levels of employees [31]. Thus, longitudinal productivity costing studies in hospital settings are necessary to inform human resource managers about the underlying long-term productivity and health impacts after exhibiting sickness presenteeism during seasonal influenza heavy workload periods.

### 4.3. Low Productivity Levels across Different Health States

This study highlights that productivity levels are overall impaired regardless of health state (healthy, presenteeism, took sick leave) in a high-intensity nursing environment and may point to other risk factors associated with presenteeism productivity loss. Siu’s Hong Kong cross-occupational study reported a similar mean presenteeism productivity loss percentage (30%) and suggested workplace job demands and stressors may have contributed to impaired employee productivity [57]. A qualitative study performed by our team on the same sample of nurses found that organizational and personal factors, such as written/unwritten rules, loyalty to colleagues and professional identity, factor into employees’ decision processes related to presenteeism behavior [58].

With regards to the non-significant difference between productivity loss of nurses who were healthy (32.6%) versus exhibiting presenteeism (33.0%), employees may have utilized self-adjustment mechanism to maintain productivity levels in face of sickness to fulfill the heavy workload demands. Researchers speculate that employees with chronic pain illness who exhibit presenteeism engage in active job crafting to maintain performance levels consistent with healthy individuals [59]. Despite maintained productivity levels among employees exhibiting presenteeism, a longitudinal study demonstrated a causal relationship between nurse presenteeism and downstream burnout, which may result in further health impacts, lower productivity and increased staff turnover [31].

### 4.4. Implications for Hospital Management and Human Resources Managers

Strategic management by human resources managers has been shown to be associated with reduced staff presenteeism behavior and positive organizational outcomes [26]. However, to achieve these outcomes, perceptions between managers and employees must be aligned. In a Chinese nationwide nurse presenteeism study, managers’ and employees’ perception on working productivity loss differed by 5% (nurses self-reported: 25.92%, managers perception: 21.01%) [56]. This study provides an important contribution for HR managers by quantifying the magnitude of self-reported presenteeism prevalence and productivity loss costs of nurse subordinates, so as to inform human resources managers of its financial impact on the hospital organization, and align managers’ perception with staff nurses on actual productivity levels when carrying out manpower scheduling exercises to tackle seasonal flu surge demands.

Moreover, our current study and previous qualitative study both point to other potential organizational and personal risk factors to be explored further [58]. Future longitudinal studies could inform how these risk factors impact presenteeism and productivity loss, so that human resources managers can modify significant organizational factors (e.g., sick leave policy, organizational commitment, job resources, supervisory support) to alleviate presenteeism behavior and improve job productivity and nurse well-being [8].

### 4.5. Limitations

Although self-rated productivity loss measures may overestimate actual productivity loss costs probably due to bias of employee’s perception on health impact on productivity, the adopted HPQ self-rated productivity measure in this study has shown high correlation with objective archival monthly supervisor work-performance ratings in a previous validation study [36].

Results presented in this study are cross-sectional in nature and do not confer causality. Differences exist in distribution methods and collection periods due to administrative issues, restricted access due to flu control measures and ethical approval constraints, despite our best efforts to match and reduce methodological and time differences in dissemination and collection processes.

Despite that data on nurse staffing changes (whether management increased staffing) during heavy workload period (winter flu surge) is not available, inpatient occupancy rate still serves as a conservative proxy for workload in such case, and our results showed a significant increase in inpatient occupancy rates during heavy workload period may be associated with higher odds of presenteeism behaviour.

## 5. Conclusions

High workload levels, as proxied by inpatient occupancy rates, are significantly associated with nurse presenteeism during seasonal influenza surge in a densely populated metropolitan city setting. Hong Kong nurse presenteeism productivity loss costs are about USD 24,096 annually, one of the highest reported costs in healthcare workforce studies worldwide. This study quantifies the magnitude of presenteeism prevalence and productivity loss costs. Reported productivity loss levels were also considerably high regardless of nurses’ health status. Future longitudinal studies should probe other modifiable organizational and personal factors that management team could adjust to reduce nurse presenteeism and increase on-the-job productivity.

## Figures and Tables

**Figure 1 ijerph-19-00769-f001:**
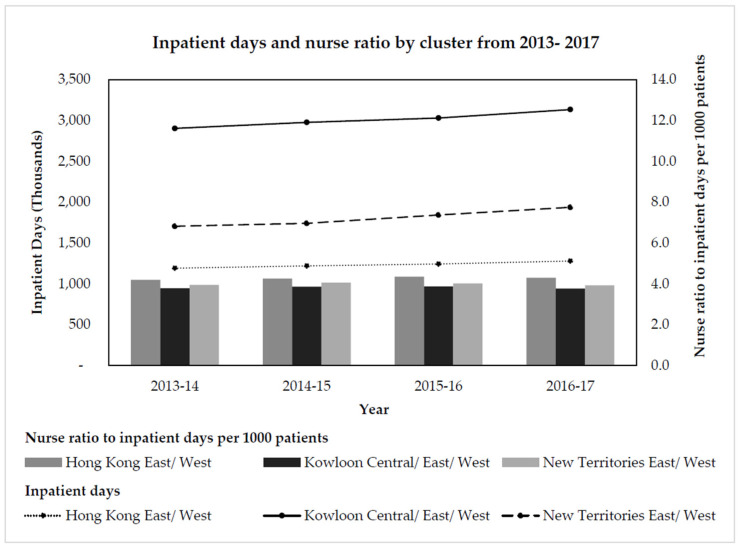
**Figure 1**. Inpatient days and nurse ratio by cluster from 2013–2017.

**Figure 2 ijerph-19-00769-f002:**
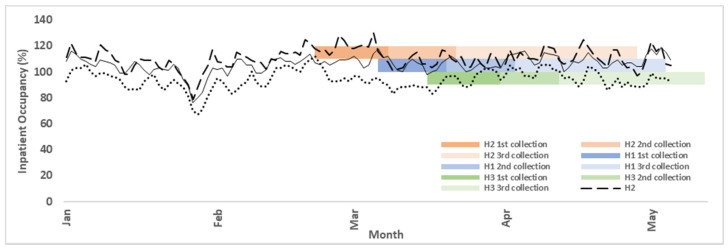
Medical inpatient bed occupancy rate for 2017 winter demand surge at three surveyed hospitals for the initial and follow up survey collection periods. Data are extracted from the Center of Health Protection (CHP)—only available for service demand surge periods (Jan 1–May 5).

**Table 1 ijerph-19-00769-t001:** Calculation of mean productivity loss percentage and costs.

Questions/Health States	How Many Times during the Last Year Have You Gone to Work When You Should Have Been on Sick Leave Due to Your Health Condition? (1—Never, 2—Once, 3—2 to 5 Times, 4—>5 Times)	Miss an Entire/Part of Work Day Because of Problems with Your Physical or Mental Health?
**Healthy**	1	0
**Sickness** **Presenteeism**	>1	0
**Sick Leave**	1,2,3,4 (anything)	>0
**Healthy and Sickness Presenteeism** Measure self-rated productivity (a) by item *“How would you rate your overall job performance on the days you worked during the past 4 weeks (28 days)? (0—worst performance, 10—best performance)”* Convert the above item to percentage productivity loss: (10 − a) × 10 = x% Discount monthly median salary by percentage productivity loss: monthly salary × x% **Sick Leave—*USD productivity loss remaining days worked*** Measure self-rated productivity (a) by item *“How would you rate your overall job performance on the days you worked during the past 4 weeks (28 days)? (0—worst performance, 10—best performance)”* Convert the above item to % productivity loss: (10 − a) × 10 = x% Measure number of days sick leave by 2 items *“ Miss an entire work day because of problems with your physical or mental health?”* and *“Miss part of a work day because of problems with your physical or mental health?”* Discount monthly median salary by % productivity loss: monthly salary / 30 days × days not on leave × x%		

**Table 2 ijerph-19-00769-t002:** Nurse characteristics by hospital.

Characteristics	H1 *n* = 824 (rr: 71.9%) *n* (%)	H2 *n* = 835 (rr: 61.1%) *n* (%)	H3 *n* = 1379 (rr: 64.3%) *n* (%)	*P* Value between Hospital Differences
**Gender**				
Male	95 (11.5)	133 (15.9)	192 (13.9)	0.034 *
Female	729 (88.5)	702 (84.1)	1187 (86.1)	
**Age Group**				
≤30	226 (28.4)	285 (35.1)	507 (38.3)	<0.001 **
31–40	196 (24.6)	235 (28.9)	265 (20.0)	
41–50	263 (33.0)	189 (23.3)	357 (26.9)	
≥51	111 (13.9)	103 (12.7)	196 (14.8)	
**Educational Qualifications**				
Certificate/diploma	72 (9.1)	65 (8.2)	111 (8.6)	<0.001 **
Associate diploma/higher diploma	47 (6.0)	40 (5.1)	176 (13.6)	
Bachelor’s degree	432 (54.8)	436 (55.1)	582 (45.1)	
Postgraduate degree	237 (30.1)	251 (31.7)	422 (32.7)	
**Rank**				
Junior staff (EN)	51 (6.2)	63 (7.5)	96 (7.0)	0.393
Junior staff (RN)	566 (68.7)	579 (69.3)	959 (69.5)	
Middle management (APN/NC)	172 (20.9)	146 (17.5)	248 (18.0)	
Senior management (WM/DOM)	35 (4.2)	47 (5.6)	76 (5.5)	
**Working Schedule**				
Shift schedule	664 (82.8)	652 (79.5)	1044 (77.6)	0.015 *
Regular schedule (9 am–6 pm)	138 (17.2)	168 (20.5)	302 (22.4)	
**Main Working Location**				
A&E	46 (5.9)	68 (8.3)	46 (3.4)	<0.001 **
GOPC/SOPC	48 (6.1)	49 (6.0)	125 (9.3)	
Medicine ^a^	309 (39.6)	384 (46.8)	536 (39.8)	
Surgery ^b^	303 (38.8)	208 (25.4)	481 (35.7)	
Others ^c^	75 (9.6)	111 (13.5)	159 (11.8)	
**Self-Rated Health (*n* (%))**				
Poor (score: 1,2)	334 (40.5)	338 (40.5)	541 (39.2)	0.011 *
Normal (score: 3)	232 (28.2)	212 (25.4)	428 (31.0)	
Good (score: 4,5)	244 (29.6)	281 (33.7)	397 (28.8)	
Missing	14 (1.7)	4 (0.5)	13 (0.9)	
**Number of Sick Leave Days (mean(sd))**	0.64 (2.13)	0.65 (3.02)	0.54 (2.01)	0.593

* *p* < 0.050, ** *p* < 0.001, ^a^ Medicine—includes medicine, geriatrics, pediatrics and intensive care unit (ICU), ^b^ Surgery—includes surgery, obstetrics, gynecology and operation theatre, ^c^ Others—includes administration, management, residential care, public health, rehabilitation, occupational health, community nursing, mental health, psychiatry, addiction treatment and others; Note: EN: enrolled nurses (EN); RN: registered nurses; APN: advanced practice nurse; NC: nurse consultant; WM: ward manager, DOM: department operations manager; H1: Hospital 1; H2: Hospital 2; H3: Hospital 3; A&E: accident and emergency department; GOPC: general out-patient clinics; SOPC: specialist outpatient clinic; rr: response rate.

**Table 3 ijerph-19-00769-t003:** Nurse self-reported presenteeism and productivity.

Nurse Self-Reported Presenteeism and Productivity	*n* = 3038 (rr: 71.9%) *n* (%)
Presenteeism Frequency	
“How many times during the last year have you gone to work when you should have been on sick leave due to your health condition?”	
Never	352 (11.8)
Once	508 (17.0)
2–5times	1440 (48.2)
>5 times	689 (23.1)
	*n* (sd)
Self-rated Productivity ^a^	6.61 (1.57)

^a^ Self-rating on job performance for past 4 weeks (28 days) (0—worst performance to 10—best performance). Note: rr: response rate.

**Table 4 ijerph-19-00769-t004:** Conventional and generalized estimating equation results on presenteeism frequency amongst nurses for hospital level characteristics.

Hospital-Level Risk Factor	Generalized Estimating Equation (GEE) Regression Adjusted OR (95% CI) ^a,b,c^	Conventional Logistic Regression Adjusted OR (95% CI) ^b^
Occupancy z-score	1.19 * (1.05–1.35)	1.18 (0.98–1.42)

* *p* < 0.050; ^a^ Nurses who are in senior management or mainly work at GOPC/SOPC or other departments (administration, management, residential care, public health, rehabilitation, occupational health, community nursing, mental health, psychiatry, addiction treatment and others) were excluded in the analysis so to measure the effect of workload on presenteeism amongst inpatient ward nurses only, ^b^ Model adjusted for hospital, age, rank, shift, health, primary working function and sick leave, ^c^ Corrected for small-sample bias; Note: GOPC: general out-patient clinics; SOPC: specialist outpatient clinic; OR: odds ratio; CI: confidence interval.

**Table 5 ijerph-19-00769-t005:** Monthly estimates on productivity loss per nurse employee by hospital.

Health States of Nurses	On-the-Job Productivity Loss (%) ^a^	Mean Sick Leave (days)	Sick Leave Costs (USD) ^b^	Productivity Loss (USD) ^b^
**Healthy**	32.6	-	-	1983
**Presenteeism**	33.0	-	-	2008
**Took Sick leave**	37.4	2.56	622	2081

^a^ Self-rated productivity for past 4 weeks (28 days) (0—worst performance to 10—best performance), mean % monthly productivity loss = (10 − a); ^b^ Median salary = USD 6083; Note: (1 USD = 7.8 HKD).

## Data Availability

The data presented in this study are available on request from the corresponding author.

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
