# Peer review of "Does Seasonal Influenza Related Hospital Occupancy Surge Impact Hospital Staff Sickness Presenteeism and Productivity Costs?"

_ijerph, 2022, doi:10.3390/ijerph19020769_

Round 1
Reviewer 1 Report
Thanks for the opportunity to review this paper. Overall, I think the manuscript deals with exciting and appropriate issues, i.e., presenteeism, productivity, nurses, influenza and hospital occupancy. I believe much more empirical research is needed. However, I do think there are some issues in this manuscript that warrant further attention. My comments follow the given structure below:
First, this paper needs a bit more detailed treatment of the recent empirical literature on presenteeism, productivity, nurses, influenza and hospital occupancy. I can see the further prospect or development of the applied theories in the literature to justify its current standing and the relation of this study. For example, I cannot see any reference of papers from 2020-21 to understand the current standing of the studied variables (e.g., presenteeism, productivity, nurses, influenza and hospital occupancy). Hence, the context and literature of the empirical analysis should be updated with 2020-papers. For example, you may see Li et al., 2019; Leal & Ferreira, 2021; Haque, 2021; Hubble & Renkiewicz, 2021; Tomonaga et al., 2021) for your further literature.
Second, the research gap is not clear enough. It is essential to understand how this study - what is examined here - offers new insights into the literature? For example, how should presenteeism, productivity, nurses, influenza, and hospital occupancy be managed effectively for higher organisational performance?
Third, the sampling and the justification of (NH1= 2145, NH2=1367 and NH3=1145) need the detail of the participants (e.g., who, why and how? context of Hong Kong?).
Fourth, when the paper considers presenteeism with workloads and productivity, there is an angle of the role of HRM or HR departments. Hence, a bit of discussion focusing on HRM overcoming presenteeism and considering the workload and higher productivity should be followed. Please see Haque (2018) in this regard.
Fifth, it requires a bit more input for data administration, collection and analysis. For example, why multilevel hierarchical logistic regression (HLM)? It requires a bit more justifications.
Finally, I found the discussion of practical implications is a bit unclear. How should practices be designed or aligned within the healthcare industry in Hong Kong? What new insight can this paper offer for the importance of presenteeism, productivity, nurses, influenza and hospital occupancy? How should presenteeism, productivity, nurses, influenza and hospital occupancy be managed? It is not easy to see clear and new takeaways here. I see the prospect, but this paper can be better with its solid practical contributions.
Good luck!
References:
Li, Y., Zhang, J., et al., (2019). The effect of presenteeism on productivity loss in nurses: the mediation of health and the moderation of general self-efficacy. Frontiers in psychology, 10, 1745.
Leal, C. C., & Ferreira, A. I. (2021). In Sickness and in Health: The Role of Housework Engagement in Work Productivity despite Presenteeism. The Spanish Journal of Psychology, 24.
Haque, A. (2021). The effect of presenteeism among Bangladeshi employees. International Journal of Productivity and Performance Management. DOI: 10.1108/IJPPM-06-2020-0305
Hubble, M. W., & Renkiewicz, G. K. (2021). Estimated Cost Effectiveness of Influenza Vaccination for Emergency Medical Services Professionals. Western Journal of Emergency Medicine, 22(6), 1317.
Tomonaga, Y., Zens, K. D., Lang, P et al., (2021). Productivity losses due to influenza and influenza-like illness in Switzerland: results of the Swiss Sentinel Surveillance Network in a non-pandemic era. Swiss Medical Weekly, (33).
Haque, A. (2018). Strategic human resource management and presenteeism: a conceptual framework to predict human resource outcomes. New Zealand Journal of Human Resources Management, 18(2), 3-18.
Author Response
Author's Reply to Reviewer:
Comment 1.1
First, this paper needs a bit more detailed treatment of the recent empirical literature on presenteeism, productivity, nurses, influenza and hospital occupancy. I can see the further prospect or development of the applied theories in the literature to justify its current standing and the relation of this study. For example, I cannot see any reference of papers from 2020-21 to understand the current standing of the studied variables (e.g., presenteeism, productivity, nurses, influenza and hospital occupancy). Hence, the context and literature of the empirical analysis should be updated with 2020-papers. For example, you may see Li et al., 2019; Leal & Ferreira, 2021; Haque, 2021; Hubble & Renkiewicz, 2021; Tomonaga et al., 2021) for your further literature.
Comment 1.2:
Second, the research gap is not clear enough. It is essential to understand how this study - what is examined here - offers new insights into the literature? For example, how should presenteeism, productivity, nurses, influenza, and hospital occupancy be managed effectively for higher organisational performance?
Response: Thank you for your comments. We have updated references (published between 2017-2021) on existing literature related to the studied variables (i.e. presenteeism, productivity, nurses, why did we choose to study influenza surge and its related hospital occupancy rate increase). Following the description of empirical literature, we have added description with regards to the research gap, how our study offers new insights to the field, and how nurse managers can benefit from our research to manage their staff more effectively for higher organizational performance.
The abstract section is now updated in the manuscript (Abstract section, lines 9-28, pg1)
The introduction section is now updated in the manuscript (Introduction section, lines 44-47 and 80-113, pg2-3)
Comment 1.3:
Third, the sampling and the justification of (NH1= 2145, NH2=1367 and NH3=1145) need the detail of the participants (e.g., who, why and how? context of Hong Kong?).
Response: Thank you for your comments. Additional context and background information on the Hong Kong public hospital cluster division, inpatient bed days and nurse to patient ratio has been included in the introduction section. Furthermore, number of beds are provided for each hospital in the methods section.
The above information provides justification for selecting the three distinct hospital types (H1, H2 and H3) within the busiest hospital cluster district with the lowest nurse to patient ratio, i.e. nurses working in this cluster face highest risk of presenteeism due to the high demands of flu surge and manpower shortage and would require additional attention from management/ HR in terms of staff scheduling, resource and manpower allocation during flu surges.
The introduction section is now updated in the manuscript (Introduction section, lines 49-67, pg 2).
The methods section is also updated in the manuscript (Methods section, lines121-136, pg3-4).
Comment 1.4:
Fourth, when the paper considers presenteeism with workloads and productivity, there is an angle of the role of HRM or HR departments. Hence, a bit of discussion focusing on HRM overcoming presenteeism and considering the workload and higher productivity should be followed. Please see Haque (2018) in this regard.
Response: Thank you for your comments. We have now factored in the angle of HRM and HR departments on staff presenteeism, workload and productivity in our introduction and discussion sections.
The introduction section is now updated in the manuscript (Introduction section, lines 105-113, pg3).
The discussion section is now updated in the manuscript (Discussion section, lines 392-409, pg 11).
Comment 1.5
Fifth, it requires a bit more input for data administration, collection and analysis. For example, why multilevel hierarchical logistic regression (HLM)? It requires a bit more justifications.
Response:
Thank you for your comments. Covariate measures and data analysis procedures are described in sections 2.3, 2.4.1 to 2.4.3, with detailed description of each individual measure scale and its origin, presenteeism prevalence calculations, justifications for using GEE model, conversion of inpatient occupancy rate to Z scores, covariates considered in the regression model with supporting literature, sample selection, handling of missing data using multiple imputation and software used for analysis.
We have now included additional information with regards to data administration, collection and analysis.
For data administration and collection, it has been previously described in section 2.2, which is now renamed “Survey administration and collection procedure” (Methods section, line 138-154, pg4)
The justification for using GEE model was previously included in the methods section is further updated ( Methods section, lines 204-210, pg 5)
Comment 1.6:
Finally, I found the discussion of practical implications is a bit unclear. How should practices be designed or aligned within the healthcare industry in Hong Kong? What new insight can this paper offer for the importance of presenteeism, productivity, nurses, influenza and hospital occupancy? How should presenteeism, productivity, nurses, influenza and hospital occupancy be managed? It is not easy to see clear and new takeaways here. I see the prospect, but this paper can be better with its solid practical contributions.
Response: Thank you for your comments, we have added section 4.4 in the discussion section on “Implications for hospital management and human resources managers” (Discussion section, lines 393 - 409, pg11)
Thank you again for the above comments.
Best Regards,
Juliana Lui
Reviewer 2 Report
Dear authors,
thank you for opportunity to review such nice work. please corect:
Abstract - please in Aim, Methods, Results form
Introduction
- text under 1.2. is not related to the subtitle. Alos, explain number of nurse per patient, bed and number of absenteezim in Hong Kongv
- Matherials
- 2.1. readers do not know nothing about tha number again, how many beds, patients, proceedure per year. also this should be in the introduction
- how and why can you connect productivity with presenteeism?
- why you chosen the season with flu? and what can be be with young nurses ( why they feel sick for work??? because of the flu? Are they vaccinated?
Results - I can not see analyses between gender and age or work place conneceted with presenteeism.
What is the theme of 4.3 paragraph? You didn't compare it it with other countries.
Although interesting manuscript, I think you made it complicated to follow and find adequate conclusion.
References - please include more references in last 10 years.
Please add implications for practice.
Author Response
Author's Reply to Reviewer
Comment 2.1:
Abstract - please in Aim, Methods, Results form
Response:
Thank you for your comments. The abstract is now updated to follow the format as suggested by reviewer 2.
The abstract section is updated (Abstract section, lines 9-28 , pg 1)
Comment 2.2:
Introduction
text under 1.2. is not related to the subtitle.
Response: Thank you for your comments. The title for this section is now updated to “1.3 Productivity loss costs, influenza and staff presenteeism”, the text is also updated to match the section subtitle (Introduction section, lines 80-113, pg2-3)
Comment 2.3:
Also, explain number of nurse per patient, bed and number of absenteeism in Hong Kong
Comment 2.4:
Methods
2.1. readers do not know nothing about the number again, how many beds, patients, procedure per year. also this should be in the introduction
Response: Thank you for your comments. Additional context and background information on the Hong Kong public hospital cluster division, inpatient bed days and nurse to patient ratio has been included in the introduction section. Furthermore, number of beds are provided for each hospital in the methods section.
The above information provides justification for selecting the three distinct hospital types (H1, H2 and H3) within the busiest hospital cluster district with the lowest nurse to patient ratio, i.e. nurses working in this cluster face highest risk of presenteeism due to the high demands of flu surge and manpower shortage and would require additional attention from management/ HR in terms of staff scheduling, resource and manpower allocation during flu surges.
The introduction section is now updated in the manuscript (Introduction section, lines 49-67, pg 2).
The methods section is also updated in the manuscript (Methods section, lines121-136, pg3-4).
Comment 2.5:
how and why can you connect productivity with presenteeism?
Response: Thank you for your comments. We have now included existing literature to support the relationship between presenteeism with productivity.
The introduction section is updated (Introduction section, lines 80-103 , pg 3)
Comment 2.6:
why you chosen the season with flu? and what can be with young nurses ( why they feel sick for work??? because of the flu? Are they vaccinated?
Comment 2.7
References - please include more references in last 10 years
Response: Thank you for your comments. We have updated references (published between 2017-2021) on existing literature related to the studied variables (i.e. presenteeism, productivity, nurses, why did we choose to study influenza surge and its related hospital occupancy rate increase). Following the description of empirical literature, we have added description with regards to the research gap, how our study offers new insights to the field, and how nurse managers can benefit from our research to manage their staff more effectively for higher organizational performance.
The abstract section is now updated in the manuscript (Abstract section, lines 9-28, pg1)
The introduction section is now updated in the manuscript (Introduction section, lines 44-47 and 80-113, pg2-3)
Comment 2.8
Results - I cannot see analyses between gender and age or work place connected with presenteeism.
Response:
Our regression model is adjusted for the covariates - hospital, age, rank, shift, health, primary working function and sick leave. For clarity purposes, we have only included the main independent variable of interest – occupancy rate in table 4, details of adjusted covariates were not shown. However, for the interest of readers, we have now included the full model parameters (including the adjusted covariates) in supplementary table 1
Comment 2.9
What is the theme of 4.3 paragraph? You didn't compare it with other countries.
Response:
The theme of section 4.3 is now clarified – “Low productivity levels across different health states”. The discussion is also updated to support this . We have also now included comparison of productivity costs and productivity loss results with other countries in sections 4.2 and 4.3 (Discussion section, lines 350-371 and 372-390, pg10-11)
Comment 2.10
Although interesting manuscript, I think you made it complicated to follow and find adequate conclusion.
Response: Thank you for your comments. We have now updated the discussion to strengthen the main themes, please kindly refer to the above response. We have also clarified our conclusion, which ties in better with our updated discussion (Conclusion section, lines 427-438, pg12)
Comment 2.11
Please add implications for practice.
Response: Thank you for your comments, we have added section 4.4 in the discussion section on “Implications for hospital management and human resources managers” (Discussion section, lines 393 - 409, pg11)
Thank you again for the above comments.
Best Regards,
Juliana Lui
Round 2
Reviewer 1 Report
Best of luck!
Author Response
We would like to express our gratitude to the editors and reviewers for their constructive and insightful comments .
We have checked on the English language style and spell check on the manuscript once more. Minor edits were made to the sentences below to improve the flow:
(Introduction section, lines 57-59, pg2):
“Nurses in the Kowloon clusters (KEC, KWC and KCC) face the highest service demands (total number of inpatient patient days) with lowest manpower (nurse ratio to inpatient days) as compared to healthcare workers in other clusters (Figure 1) [10, 11].”
Best Regards,
Juliana Lui
Reviewer 2 Report
Dear authors,
sorry for waiting, here are comments.
Manuscript is improved and easier to follow and read.
Please correct line 59.
In conclusion, I would leave ot this part: "to inform human resources managers of its financial impact on the hos-432 pital organization, and encourages nurse managers to factor in presenteeism-related 433 productivity losses when carrying out manpower scheduling exercises" as you didn't mention it in the aim.
Wish you happy New Year with more manuscripts like this.
